# A Fully Integrated RFID Reader SoC

**DOI:** 10.3390/mi14091691

**Published:** 2023-08-29

**Authors:** Jian-Guo Hu, Wen-Zhuo Mei, Jin Wu, Jia-Wei Li, De-Ming Wang

**Affiliations:** 1School of Microelectronics Science and Technology, Sun Yat-sen University, Zhuhai 519082, China; hujguo@mail.sysu.edu.cn (J.-G.H.); 13071204690@163.com (W.-Z.M.); 2School of Electronics and Information Engineering, South China Normal University, Foshan 528225, China; 3Development Research Institute of Guangzhou Smart City, Guangzhou 510805, China; wj@gz-smartcity.org.cn

**Keywords:** RFID reader, Cortex M0, digital baseband, SoC

## Abstract

The traditional RFID reader module relies on a discrete original design. This design integrates a microcontroller, high-frequency RFID reader IC and other multiple chips onto a PCB board, leading to bottlenecks in cost, power consumption, stability and reliability. To align with the trend towards high integration, miniaturization and low power consumption in RFID reader, this paper introduces a fully integrated RFID Reader SoC. The SoC employs the open-source Cortex-M0 core to integrate the RF transceiver, analog circuits, baseband protocol processing, memory and interface circuits into one chip. It’s compatible with ISO/IEC 14443 A-type and B-type and ISO/IEC 15693 transmission protocols and rates. Manufactured using a 0.18 μm process, the chip is compatible with multiple standards. The optimized design of the digital baseband control circuit results in a chip area of only 11.95 mm${}^{2}$ offering clear advantages in both area and integration compared to similar work.

## 1. Introduction

With the advancement of ultra-large-scale integrated circuits, the Internet of Things (IoT) and information security technologies [1], RFID and NFC technologies have gained attention for their unique features [2]. When integrated with the Internet and communication technology, the RFID system has proven effective in access control, supply chain management, target tracking and other applications [3,4]. These systems have potentially permeated all aspects of human life, becoming a foundational technology for the future of information. They have been hailed as the 21st century’s most innovative and promising information technology [5,6].

An RFID system consists of three main components: reader, transponder and application software system [7,8]. The reader sends a specific frequency of radio wave energy to the transponder, activating it to transmit its internal data. The reader then receives, interprets and forwards this data to the application program for processing [9,10]. Various standards govern different operating frequencies and data rates, with High Frequency (HF) (13.56 MHz) and Ultra High Frequency (UHF) (US: 902–928 MHz; China: 840–845 MHz and 920–925 MHz; Europe: 865–868 MHz) being the most common [11]. HF technology is limited in terms of communication distance and rate, but it offers the advantages of cost-effectiveness, energy efficiency, robust penetration of non-metallic objects, and immunity to ambient noise and electromagnetic interference (EMI), compared with UHF, which limits immunity to EMI [12,13]. So compared with RFID tags operating in other frequency bands, HF tags have found broad applications and command the largest production volume [14,15]. Typically a reader can handle hundreds of tags.However, due to different standards of HF RFID tags on the market, there is a growing interest in a dedicated RFID reader that is compatible with all RFID tags [16,17]. To address this trend, this paper focuses on the design and implementation of HF multi-standard RFID reader.

The primary challenges in designing high-frequency multi-standard RFID reader include achieving high integration, miniaturization and low power consumption [18]. Historically, designers used discrete components, integrating RF transceiver chips, CPU, DSP and other units from various companies onto a single PCB board [19,20]. While this approach allowed for the utilization of best-in-class products, it also led to high costs, complex circuit layouts and limitations in miniaturization and power reduction [21,22]. This paper introduces a fully integrated SoC for RFID reader to address these challenges. The main contributions are as follows:

1. The proposed SoC chip achieves fully integrates RF analog circuits, baseband protocol processing units, microprocessors, memory and interface circuits into a singular chip. This integration significantly minimizes the area of the reader’s PCB board and simplifies the layout and wiring complexities, aligning with the contemporary development trends in RFID reader technology.

2. The internal architecture of the SoC chip has been designed with optimisation. By directly connecting the microprocessor and baseband protocol processing unit via the APB bus and opting for RAM instead of internal flash, the design reduces the number of control circuits substantially. This results in a marked reduction in the chip’s internal area.

3. The HF RFID reader SoC designed herein demonstrates compatibility with several communication standards, including ISO/IEC 14443 Type A and Type B and ISO/IEC 15693. Additionally, it supports a wide range of transmission rates (from 106/212/424/848 kbit/s to 1.54 kbit/s to 53 kbit/s), thereby enhancing its applicability across various scenarios and requirements.

The paper is organized as follows: Section 2 outlines the overall architecture of the RFID reader SoC. Section 3, Section 4 and Section 5 detail the full SoC implementation. Section 6 covers the software design, and Section 7 presents the test results and conclusions.

## 2. System Consideration and Architecture

### 2.1. Review of Standard

The primary protocols for 13.56 MHz RFID systems include ISO/IEC 14443 Type A and Type B and ISO/IEC 15693. Specific parameters for these protocols are detailed in Table 1.

These protocols do not operate on a full-duplex communication mechanism; instead, they employ a half-duplex communication paradigm, where the reader transmits information first, followed by a response from the tag. The communication thereby follows an alternating pattern between the two entities.The protocol standard may be bifurcated into two distinct segments: the physical layer structure and the tag identification layer structure. While the physical layer is responsible for aspects such as data checksum encoding and decoding, packet formatting and data transfer rates, the tag identification layer primarily focuses on the command structures facilitating reader interaction with the tags for reading and writing operations.

The reader device developed in this study is designed to be compatible with the aforementioned three protocols. Control over the associated parameters and commands is managed through the internal functioning of the transceiver SoC.

### 2.2. Hardware and Software Co-Design

This paper presents a SoC design for an RFID reader, with the primary goal of establishing interactive communication between the reader and the tag system, in compliance with the standards of the communication protocols. As outlined in the previous section, these three communication protocols are organized into two distinct layers: the physical layer and the tag recognition layer. The physical layer’s main function is to implement baseband coding, create data packets that conform to the protocol standards and then transmit them serially. On the other hand, the tag recognition layer of the protocol encompasses aspects such as the command system, the transitions between the reader and the corresponding states of the tag and the initiation of communication or responses. In traditional read-write operations, the microprocessor sends relevant commands to the radio frequency chip through the communication serial port. A state transition controller is designed inside the RF chip to receive and analyze data from the microprocessor. It then instructs the codec module and analog front end to perform the next step of the operation, completing the tag recognition layer. This approach does not fully utilize the microprocessor’s performance and leads to complex control circuit design. Moreover, the serial communication mechanism imposes a speed bottleneck.

To fully utilize the MCU and reduce the chip area, the final design in this paper divides the hardware and software as follows: The baseband protocol unit is implemented in hardware and the processor is connected through a high-speed bus to meet interaction speed and timing requirements. The baseband protocol unit accepts control signals and data from the processor to complete the physical layer functions, including codecs, packet format and data transfer rate. It then transmits the baseband signal to the RF analog front end part, sending data in a format that meets protocol standards. The tag recognition layer and overall system control are mainly achieved through the development and writing of software programs by the processor.

### 2.3. Proposed Reader Schema

The architecture of the proposed system is depicted in Figure 1. The SoC employs ARM’s Cortex-M0 as the CPU and incorporates both an AHB-Lite bus and an APB bus. The kernel supports debugging through the JTAG interface for external devices. Additionally, the Power Management Unit (PMU) module oversees the power-up and power-down processes of the kernel and manages the system clock. The reset control module generates essential system reset signals, such as power-on reset, AHB system reset, and APB system reset. An internal RC oscillator is included in the system, generating a 100 KHz clock as the system’s low-frequency clock for low-power mode. The AHB bus hosts various components, including a system controller, a 4 KB ROM, a 64 KB Flash, three GPIOs and a ROM table.

The APB bus in the system hosts an independent watchdog, two serial ports, two timers and an RFID reader. The protocol processing circuitry within the RFID reader is capable of encoding and decoding multiple standard packages, such as ISO/IEC 14443 Type A, ISO/IEC 14443 Type B and ISO/IEC 15693. The system utilizes an external crystal oscillator operating at 27.12 MHz. Through an internal frequency divider, a 13.56 MHz system frequency is derived, while the 27.12 MHz clock is employed in both the Flash interface and the RFID reader.

For the consideration of low power consumption, the actual circuit does not have flash because the SoC chip in the Flash program memory control is highly complex; its various work control work has strict control of the timing, and the control of Flash program memory will consume many hardware circuit resources, resulting in increased power consumption. In this paper, instead of a piece of RAM, the power-on reset before the hardware design, the external piece of the SPI FLASH program transfer to the RAM, an initialization process, in the reset, the function of the RAM and the function of FLASH a no different. And when the user program is downloaded via rebootloader, the location of the download storage is the external SPI FLASH. The power-on program transfer module is flash2ram.v; the idea is actually to read the flash from address 0 and read the 64 KB program and write it into the RAM. Using the settings to copy the Flash program in the Flash program memory to SRAM/RAM and then running and controlling. Normally, the Flash program memory control of the various control work usually occupies a very short running time, especially read data control work will also be a very short running time of the process. In this case, power consumption can be reduced by directly turning off the peripheral clock and power supply of the entire Flash program memory after the copy operation of copying the Flash program from the Flash program memory to RAM is completed.

## 3. Reader Hardware System Design

The hardware system of this paper is developed using a top-down design approach, and its structure is depicted in Figure 2. The design process begins with the creation of a general-purpose processor, enabling standard functionalities such as serial port transmission, timer timing and GPIO control. Subsequently, the RF function module is modelled and treated as a peripheral, mounted on the general-purpose processor’s APB bus. This configuration allows the user to access the RFID function module through software calls via the general-purpose processor, thereby completing the construction of the SoC chip.This section provides an in-depth examination of the working mechanism and implementation details of each core module within the SoC. The reader module will be highlighted in the subsequent section.

### 3.1. Cortex-M0 Processor

Among the available Cortex series processors, the Cortex-M series is optimized for microcontroller-based applications. The Cortex-M0 processor, one of the smallest Arm processors available, boasts an exceptionally small silicon area, low power consumption and minimal code footprint. This is attributed to the Thumb2 subset of the Thumb ISA. The processor’s minimal footprint ensures reduced code size and memory usage. Its 3-stage pipeline provides the parallelism required by the application while minimizing instruction overhead. While a large number of pipeline stages can be used for high throughput, they require more independent instructions. Since our application neither requires high throughput nor the complexity of scheduling independent instructions, the ARM Cortex-M0 is best suited for our application.

### 3.2. Power Management Unit

To further minimize the power consumption of the SoC chip while leveraging the low-power mode of the Cortex-M0 integrated into the SoC system, the Power Management Unit (PMU) is bifurcated into two segments. The first segment is responsible for the kernel’s power management, both up and down, and the second segment controls the system clock. The primary running clock for all peripherals is derived from the free-running clock FCLK. The running clocks for peripheral components can be toggled on or off via the register control of the sysctrl module. Other clocks are contingent on the system’s low-power mode, sleep mode and deep-sleep mode. In sleep mode, only the kernel clock is halted, and HCLK is deactivated, causing peripherals dependent on HCLK to cease operation while other clocks maintain their original state. In deep sleep mode, FCLK is switched to a low-speed clock of 100 KHz and peripheral functions related to HCLK are halted, with other peripherals relying on the register control switches of the sysctrl module.

### 3.3. Memory

To fulfil the fundamental functions of the processor, this paper introduces the design of three types of general-purpose memory: Read-Only Memory (ROM), Flash Memory (FLASH) and Random Access Memory (RAM).

ROM: Typically utilized to store the bootloader program, ROM offers two startup methods: flash main memory startup and ROM startup. The selection between these methods is controlled by the external BOOT pin. If the BOOT pin is set to 0, the system will boot from the flash main memory following a reset; otherwise, it will boot from the ROM.

FLASH: Serving as the principal repository for user programs, FLASH offers a storage capacity superior to ROM. It is instrumental during the bootloader phase, enabling the transition to the user-defined program. The flash interface within the design operates across two clock domains at frequencies of 13.56 MHz and 27.12 MHz. Given the rapid 25 ns access time for reading and the constraint of single-byte data retrieval per read, the 27.12 MHz frequency is employed to expedite the kernel’s data retrieval, thereby enhancing the SoC’s overall performance. All components, barring the read access interface, function at 13.56 MHz.

RAM: Allocated to furnish resources during the Cortex-M0 kernel’s control and logical operations, RAM’s integration requires minimal human intervention. The design necessitates the mounting of RAM, of an appropriately gauged size, onto the AHB bus, in accordance with the specifications of the project.

### 3.4. Peripherals

For each SoC, peripherals are needed to cooperate with the processor to realize the function. This paper focuses on the basic functions of the reader, choosing to mount the corresponding peripherals.

(1) GPIO: As the interface resource extension of the SoC, there are 3 groups of GPIOs, which are mounted on the AHB bus, named GPIO0, GPIO1 and GPIO2, among which GPIO0 and GPIO1 are open to the user and GPIO2 is used for internal control of special functions. GPIO0 is a general-purpose GPIO without multiplexing and GPIO1 is a multiplexed GPIO;each pin can be used as a general IO or special function. GPIO1 is a multiplexed GPIO, each pin can be used as general IO or special function pin.

(2) UART: The Cortex-M0 does not have a memory management unit (MMU), so it is not possible to run a traditional operating system on it. Although a real-time operating system (RTOS) can be used, it requires more memory to run and is not needed in our application. In this paper, UART is used to allow program changes in SoC with less overhead mounted on APB bus.

(3) TIMER: The communication between the reader and the tag must meet the link timing requirements specified by the protocol. The timing requirements use an internal timer to achieve, when more than a certain period of time, the internal timer generates an interrupt, the reader enters a new implementation of the process and no longer waits for the tag response.

(4) Watchdog: To prevent the system from stalling for a long time or entering a non-responsive state. If the system fails or deadlocks and can not run, the watchdog will be triggered and take appropriate measures to help ensure the stability of the system to avoid possible data loss or system crash. In this paper, mounting the watchdog on the Cortex-M0 can provide a monitoring and protection mechanism to ensure the reliability of the reader chip and take appropriate action in case of system anomaly.

(5) JTAG: JTAG debugging interface is the more widely used debugging interface for chip debugging at present. It generally has two important applications, one is to replace the processor to access the devices on the bus, so that it can view each device and the other is to connect with the development interface of the processor to realize the debugging functions such as breakpoints, single-steps and tracing. In debug mode, the JTAG debug module suspends the processor and replaces it as the primary controller to operate and diagnose peripherals, update ROM/Flash and read and modify the processor’s internal state. Such operations are necessary during the debugging and simulation phase of the design.

## 4. Digital Baseband

The digital baseband, compliant with ISO/IEC 14443 Type A, ISO/IEC 14443 Type B and ISO/IEC 15693 standards, encompasses the modules depicted in Figure 3.

This baseband interfaces with the system through the APB bus. The design of this custom IP core’s interface adheres to the APB interface protocol, primarily comprising bus slave device signals and protocol processing unit signal definitions. To facilitate the internal bus in reading and transmitting data to the bus slave, a dual-port RAM has been integrated into the design. This serves as a buffer for data from both the AMBA bus and the analog front end, thereby negating the necessity for intricate interfaces to implement the AMBA protocol. Consequently, this approach simplifies the system design and reduces its complexity.

In the bidirectional communication process between the reader and the tag, the microcontroller functions as a controller, issuing commands. The timer module processes incoming CLK signals, dividing them into suitable frequencies, while the transmitter module encodes the data. In conjunction with the CRC module, the transmitter generates a valid response frame. Simultaneously, the receiving module cooperates with the CRC module to verify the legitimacy of the incoming frame and to extract pertinent command data. Provided that the frame is validated, the processor oversees the categorization and execution of the extracted command data. Should the command pertain to authentication, the AUXCAL authentication module comes into play. It creates a pseudo-random number, encrypts it utilizing the LFSR structure and generates the key for re-transmission. Subsequently, the transmitter module encrypts the data and fabricates a valid response frame.

In addition, to accommodate various standards, the AFE transceiver must dynamically adjust its transmit power to the tag in alignment with the specific standards and distinct identification ranges. Within the digital baseband, an AFE configuration module has been implemented to furnish an extensive array of control signals to both the RF transmitter and receiver circuitry.

## 5. Analog Front End

The AFE in the multi-standard RFID reader transceiver, as illustrated in Figure 4, must fulfill the following functional aspects:

1. Generate high-frequency transmit power to activate the tag chip and supply it with the necessary energy.

2. Modulate the transmit signal, which is then conveyed through the antenna and transmitted after modulation.

3. Accurately receive and demodulate the high-frequency signals emanating from the tag. It consists of two parts, the transmitter and the receiver. As shown in Table 1, the 13.56 MHz RFID system uses different ASK modulation indexes. Specifically, the 14443A standard transmits data from the reader to the tag at a 100% modulation index, whereas the ISO 14443B standard utilizes a 10% ASK modulation. To facilitate multi-standard support, distinct modulation indexes must be regulated within the ASK modulator in the transmitter (TX). Moreover, the multi-standard transceiver is designed to adjust its transmit power to the tag, contingent on the standard and the identification ranges. This adaptability is achieved through digital control bits, TXIndex and TXPower, which are the output signals of two sets of 6-bit digital circuit registers. Here, TXIndex configures the modulation depth, and TXPower determines the transmit power.

The transmitter, in its final configuration, is externally linked to the AFE configuration module within the digital baseband. This transmitter (TX) is composed of specific circuits designed for the selection of modulation index, control of read/write range and generation of Amplitude Shift Keying (ASK) modulation signals. Concurrently, the receiver encompasses a switched-capacitor sampling circuit, a variable gain amplifier (VGA), a BPF and a comparator. The external configuration is further augmented with matching circuitry and an antenna coil, ensuring optimal performance.

## 6. Software Design

The software development related to this SoC includes two aspects, namely the startup program and the application program. In this section, each of them is elaborated separately and a conflict prevention method based on Type A conflict bits is proposed as a reference for the subsequent design.

### 6.1. System Startup Procedures

The bootloader serves as a critical component in the initialization process of hardware devices, facilitating the establishment of memory space mapping. Upon successful execution of the bootloader, both the system’s hardware and software are transitioned into an optimal state, poised for subsequent system operations. Within the context of this design, the system’s start address is designated as 0000000000 and the CPU reset vector’s initial address is set at 0 × 100, marking the point from which the processor commences its initialization. The concluding phase of the startup sequence permits the program’s main memory to initiate at the terminal point of the interrupt processor program address space, continuing until memory becomes accessible.

### 6.2. Application

The application’s primary function is to facilitate device testing and oversee the reader’s baseband processing. This is achieved through the reader protocol file, which encompasses the initialization of the baseband processing module, along with the procedures for information transmission and reception. Specifically, the protocol file consists of the header file reader.h and the source file reader.c. Within the header file, the register offset address of the READER is defined, along with the macro definitions for register initialization configurations and the declarations for functions contained in the source file. The offset address definitions are categorized into several registers, including Instruction Registers, Enable Registers, Send Registers, Receive Registers and Execution Status Registers. Among these, the Enable Registers are further divided into transmit link enable and receive link enable, while the Status Registers encapsulate select status, tag collision, tag no conflict and no response.

### 6.3. Conflict Prevention Method for TypeA Based Conflict Bits

The existing anti-collision algorithm applied to Type A utilizes a binary tree search algorithm. In the event of a conflict, it is necessary to assign a value to the conflict bit and then transmit it to the card with the received UID. This approach has a significant drawback, namely the extended time required for identification. Since the card’s UID for read-write is unknown to the reader, assigning a value of 0 or 1 to the conflict bit does not make a difference. Therefore, it directly determines that where there is a conflict, the conflict bit for the Select card is set to 1 and a card can still be selected. To address this, the paper proposes a Type A conflict prevention method based on the conflict bit in the software design. By introducing an ANTI command in conjunction with the original SEL, the amount of data sent is reduced, thereby minimizing the data that the card must evaluate and consequently decreasing the detection time. The steps are as follows:

1. Card Reception of Commands: The card receives the SEL + NVB command sent by the reader and sequentially transmits UID data to the reader. The reader then receives the UID data and identifies whether there are conflicting UID data. If a conflict is detected, the UID conflict bit T + 1 is identified and the value of T + 1 is written into the lower 7 bytes of ANTI.

2. Reader Feedback: The reader sends the SEL+ANTI anti-collision command to the card, informing the card that the lower 7 bytes of ANTI correspond to the T + 1 bit of UID data.

3. Card Response: The card evaluates the SEL+ANTI anti-collision command on T + 1 to determine whether T + 1 is set to 1. If so, it sends the UID from the T + 1 bit; otherwise, it enters the idle state. If not, the reception of UID data is completed. The flow is depicted in Figure 5.

## 7. Measurement Results and Analysis

### 7.1. Chip Layout

Finally, we use the 180 nm CMOS process to make the fully integrated SoC reader we designed into a chip and its chip layout is shown in the Figure 6 below, which is a highly integrated chip that integrates RF analog circuit, single-chip microcomputer, protocol processing logic and memory, with an area of 2617 μm × 4566 μm. Table 2 summarizes the parameters of the SoC and compares them with other similar works, reference [23] synthesizes UHF and HF communication protocols, literature [24,25] and this paper work is similar are designing a multi-standard highly integrated HF RFID reader SoC, compared to this paper’s area is more advantageous, literature [26] designs a single standard HF RFID reader SoC, this paper is compatible with more protocols in the case of a slightly smaller area From the table, it can be seen that the proposed reader SoC has the advantages of high integration and small area.

### 7.2. RFID Reader Module

The chip was packaged using the SIP process and integrated into a reader module with peripheral circuits. Additionally, the design pattern of a traditional reader was employed to create a comparative reader, allowing for a direct contrast with our work and highlighting our contributions. As illustrated in Figure 7, the design presented in this paper not only minimizes the need for basic circuit components such as resistors and capacitors but also eliminates the requirement for an MCU. This streamlined approach leads to a more precise PCB layout, effectively enhancing both the layout and wiring. Furthermore, this highly integrated design contributes to a reduction in both the cost and complexity of the system, demonstrating the efficiency and innovation of our approach.

### 7.3. Testing of Communication Protocols

Communication tests were conducted between the reader SoC and commercial tags, adhering to the standard commands of ISO/IEC 14443 Type A, Type B and 15693. These tests were designed to verify the chip’s ability to realize the fundamental functions as stipulated by the protocols. The testing procedure primarily focused on observing the waveforms and demodulated signals on the antenna. Specifically, the lower yellow waveform represented the transmit waveform originating from the reader, while the upper red waveform depicted the reply waveform from the tag. Both the top and bottom displays showcased the waveforms post-amplification, allowing for an assessment of their compliance with the protocol coding rules. The following sections provide a detailed overview of the testing process for each protocol:

#### 7.3.1. Type A Protocol

Before establishing a communication link between the reader chip and the tag chip, the reader initiates the card-seeking mode by sending the initialization command REQA, controlled by the microcontroller. Upon receiving this command, the tag responds with the ATQA command. When the reader detects multiple nearby tag chips responding to the initialization command, it enters the anti-collision phase. During this phase, the reader transmits an anti-collision command, Anticollision, to identify the appropriate tag chip within the vicinity. The tag, in turn, acknowledges the anti-collision command by transmitting its UID to the reader. Following the completion of the anti-collision phase, the reader selects the target tag chip for communication by sending the Select command. Once the connection is established, the target tag replies with the SAK command. The final result is depicted in Figure 8. The reader employs 100% ASK modulation and modified Miller coding to send the REQA, Anticollision and Select commands while utilizing Manchester coding and OOK modulation to receive the ATQA, UID and SAK responses.

#### 7.3.2. Type B protocol

In the communication process conforming to the Type B protocol, the reader initiates the configuration by transmitting the Initialize REQB command. Upon receiving this command, the tag responds with an ATQB response. Subsequently, the reader sends the ATTRIB command, controlling the card’s PUPI and protocol setup parameters, to select and activate the card. The card acknowledges receipt of the ATTRIB command by returning a response that confirms its activation. Throughout this communication sequence, the reader employs 10% ASK modulation and NRZ encoding to send the REQB and ATTRIB commands. It then utilizes NRZ-I encoding and BPSK load modulation to receive the ATQB and Answer ATTRIB responses. The result of this communication process is illustrated in Figure 9.

#### 7.3.3. 15693 Protocol

In the communication process, the card receives an inventory request and subsequently returns an inventory response containing the card’s UID. The reader, upon receiving the card’s response, transmits a select command to engage a specific card for operation. During this communication sequence, the reader employs 10% ASK modulation and PPM encoding to send the inventory command. It then utilizes Manchester encoding and load modulation (with two subcarriers) to receive the UID response from the card. The result of this communication process, conforming to the Type C protocol, is depicted in Figure 10.

The test results, as depicted in the corresponding figure, confirm that all received data were successfully demodulated, thereby substantiating the capability of the chip designed in this study to effectively manage both sending and receiving communications. Furthermore, an extensive series of transmission tests were conducted, encompassing the activation of various protocols and high-speed communication at rates of 212, 424 and 848 kbps. Notably, no instances of card reading failure were detected throughout these tests. This collective evidence further underscores the stability and reliability of the chip’s design, affirming its potential applicability in real-world scenarios.

## 8. Conclusions

This paper presents the design and analysis of a high-frequency RFID reader SoC that leverages the Cortex-M0 core. The implementation is comprehensively explored across three primary architectural components: the integrated MCU, the RF analog circuit and the chip digital baseband circuit. By achieving a highly integrated multi-standard reader, the design not only effectively minimizes the complexity of the reader module but also optimizes the chip area. These advancements contribute to a valuable reference framework that may guide and inspire future research and development endeavours in the field of RFID technology. 

## Figures and Tables

**Figure 1 micromachines-14-01691-f001:**
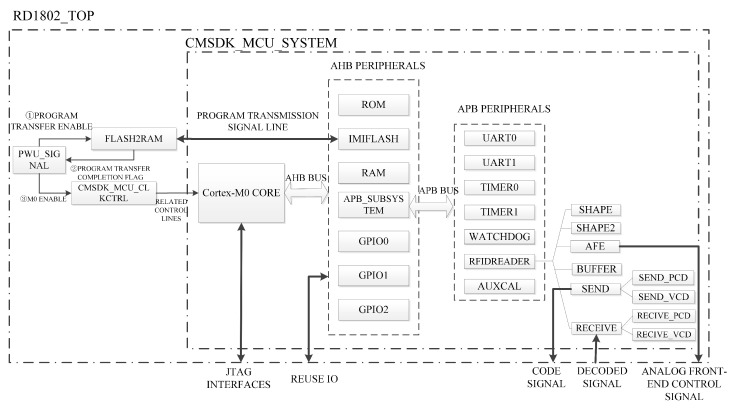
Proposed system architecture of the RFID reader SoC.

**Figure 2 micromachines-14-01691-f002:**
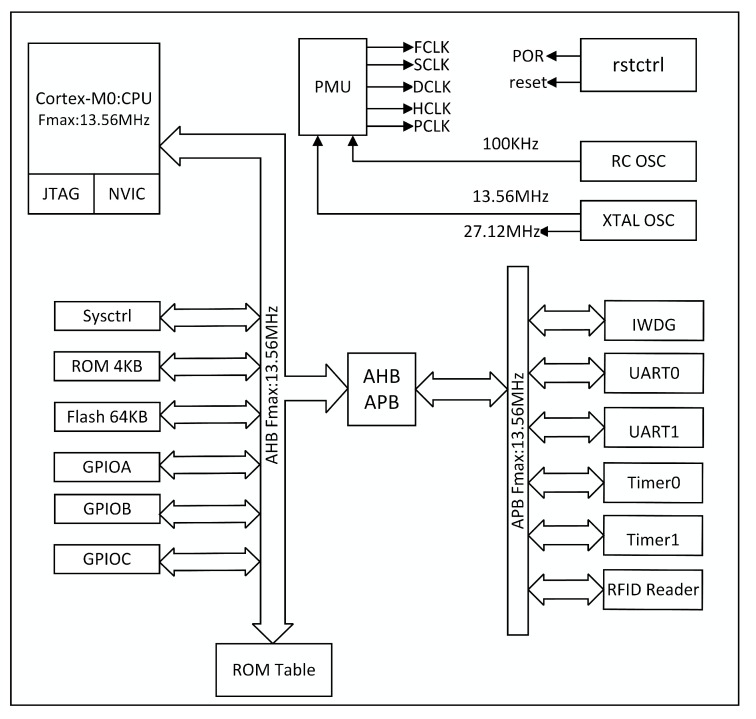
Hardware system block diagram.

**Figure 3 micromachines-14-01691-f003:**
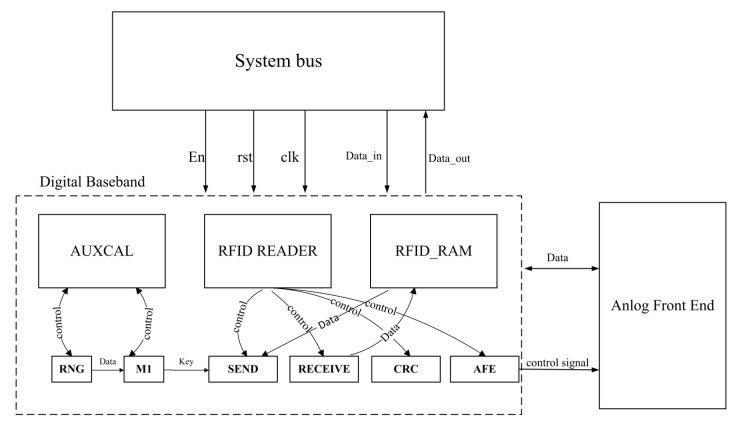
The architecture of digital baseband.

**Figure 4 micromachines-14-01691-f004:**
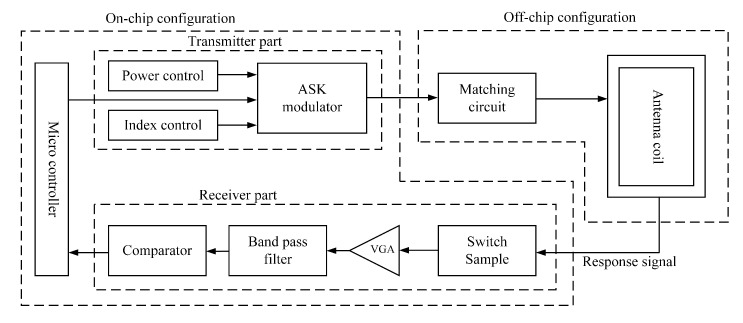
The architecture of Analog front end.

**Figure 5 micromachines-14-01691-f005:**
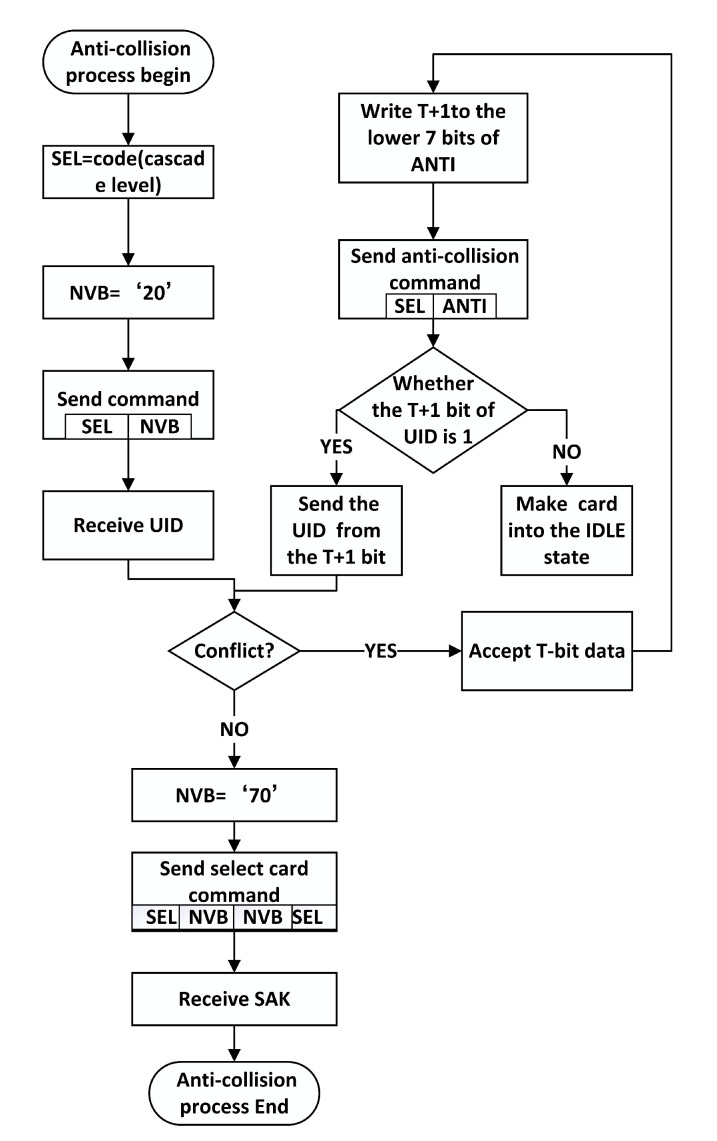
Chart of anti-collision algorithm based on type A collision bits.

**Figure 6 micromachines-14-01691-f006:**
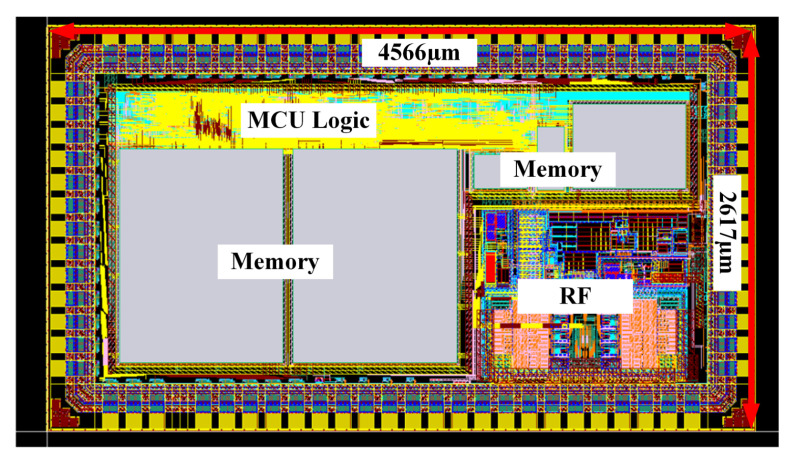
Fully integrated SoC chip layout.

**Figure 7 micromachines-14-01691-f007:**
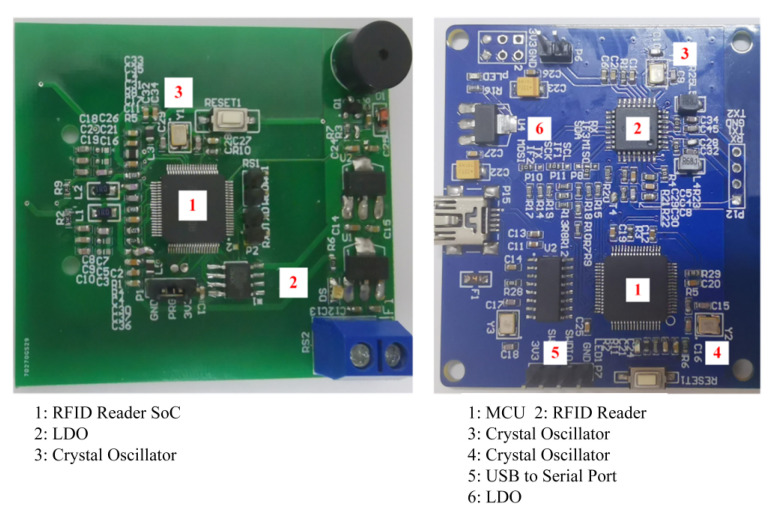
RFID reader module.

**Figure 8 micromachines-14-01691-f008:**
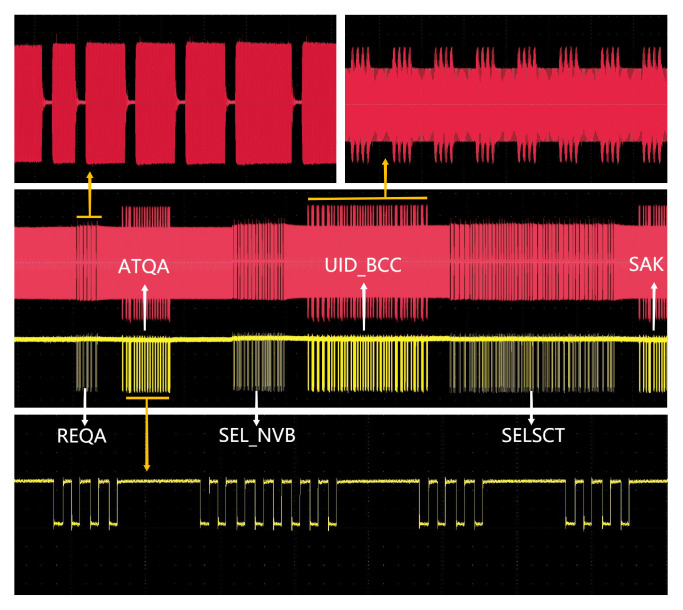
Measured transaction process of Type A protocol.

**Figure 9 micromachines-14-01691-f009:**
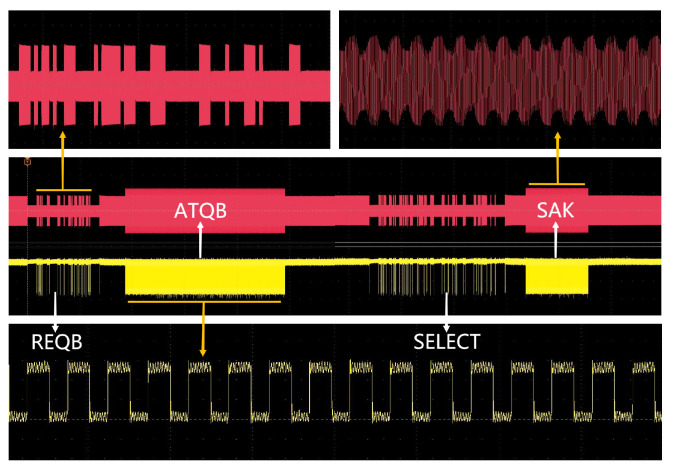
Measured transaction process of Type B protocol.

**Figure 10 micromachines-14-01691-f010:**
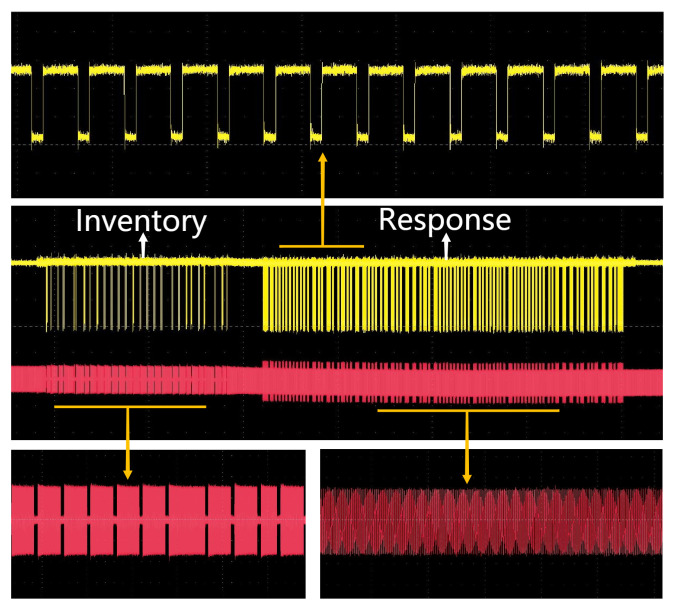
Measured transaction process of 15693 protocol.

**Table 1 micromachines-14-01691-t001:** Summary of the multi-standard 13.56 MHz RFID reader.

Standards	ISO/IEC 14443 TypeA	ISO/IEC 14443 TypeB	ISO/IEC 15693
Carrier frequency	13.56 MHz ± 7 KHz		
Subcarrier frequency	847.5 KHz		423.75/484.27 KHz
Modulation index	ASK:30–100%	ASK:8–20%	ASK: 100% or 10–30%
Magnetic field	1.5-7.5 A/m		0.15–5 A/m
Load modulation	>20/H${}^{0.5}$ mV		>10 mV
Data rate	106/212/424/848 kbits		6.62–52.97 kbit/s
Data coding (TX)	Modified Miller	NRZ-L	PPM(1/256, 1/4)
Data coding (RX)	Manchester, NRZ-L	NRZ-L	Manchester
Anti-collision	Bit oriented anti-collision	Timeslot	Bit oriented or Timeslot
Distance	0–10 cm		Up to 1 m

**Table 2 micromachines-14-01691-t002:** Performance summary OF the RFID reader SoC and comparison with the related work.

	[23]	[24]	[25]	[26]	[This Work]
Technology	90 nm	0.18 μm	0.18 μm	40 nm	0.18 μm
MCU	/	32-bit	32-bit	32-bit SC100	32-bit cortex M0
Supported Standard	ISO/IEC 14443-A ISO/IEC 15693 ISO/IEC 18092 ISO 18000-6C EPC Class1 G2	ISO/IEC 14443-A ISO/IEC 14443-B ISO/IEC 15693	ISO/IEC 14443-A ISO/IEC 14443-B ISO/IEC 15693	ISO/IEC 14443-A	ISO/IEC 14443-A ISO/IEC 14443-B ISO/IEC 15693
Data rate	106/212 kbit/s 1.54 kbit/s to 26 kbit/s	/	106 kbits 1.54 kbit/s to 26 kbit/s	/	106/212/424/848 kbit/s 1.54 kbit/s to 53 kbit/s
Data Coding	/	Modified Miller, Manchester NRZ, PPM	Modified Miller, Manchester NRZ, PPM	Modified Miller, Manchester	Modified Miller, Manchester NRZ, PPM
Chip area	13.68 mm${}^{2}$	18 mm${}^{2}$	25 mm${}^{2}$	8.95 mm${}^{2}$	11.95 mm${}^{2}$

## Data Availability

Not applicable.

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
