# Peer review of "A Fully Integrated RFID Reader SoC"

_micromachines, 2023, doi:10.3390/mi14091691_

Round 1

Reviewer 1 Report

1.       Author corrects the micrometre symbol in Abstract part.

2.       In the Introduction part reference is wrongly written. Write the reference number before the full stop.

3.       Author write properly the Table1 and Table 2 heading.

4.       Increase the font size of Figure 2 and Figure 6 block elements so that it will be clearly visible.

5.       Heading first letter must be capital so correct the: 6. software design; 6.2. application, 7.1. hardware and software co-design.

6.       Author is mentioned about hardware implementation, but the point is not clearly understandable in entire paper.

7.       Paper is writing very poor and grammatically many errors.

8.       In heading Measurement results and analysis, the first paragraph is not having proper meaning. Sentence is not connected properly. The micrometre is not writing properly.

9.       In 7.1. hardware and software co-design part author mentioned: so this paper’s final design of the hardware and software departments are shown in the following figure 7 figure 8, which is meaningless.

1.   Author writing wrongly the order of figure number. Author mentioned Figure 6 than Figure 8 after that Figure 7 and Figure 8. Author mentioned Figure 7 is hardware but as per paper it is Figure 9.

1.   Author not described the information about the key figures as Figure 9, Figure 10, Figure 11, and Figure 12 in entire paper.

1.   In chip layout mentioned the layer information. Add the description of each part that is visible in the cross-sectional picture.

1.   Figure 9 is RFID reader module, but the information is missing about it. Mentioned also it is fabricated by own or author is using for own application purpose.

1.   No meaning is reflected from Figure 10 and Figure 11, explain the result properly and increase the visibility of text in figure.

1.   Author is keeping same figure in two places i. e. Figure 1 and Figure 7 i. e. System block diagram, how it is possible.

1.   Make a comparison table of your results with the literature work.

Paper is writing very poor and grammatically many errors.

Reviewer 2 Report

This is an interesting work, however, the format of the manuscript and the writing style are needed to be improved significantly. There are several punctuation errors (commas, space, full stops, etc., are missing). Also, it can be seen that many sentences/paragraphs start with a small letter. Hence, the manuscript should be proofread carefully.

The other issue in the paper is that the novelty of the work is not presented clearly, and it is not clear what is the main contribution of this work compared to any standard system on chip working as a transceiver. 

Other suggestions:

Fig.7 is not clear and does not provide any valuable information. You may annotate the figure with the important elements of the design. Please label the components in Fig. 8., especially the designed chip.

Similarly, Figs. 9 and 10 are needed to be improved. There is no label for the plots (x and y axis), and also it is needed to specify the signal for each section of the figure. The explanation in the text should also be consistent with the figures and explain the details.

The manuscript is needed to be proofread carefully. 

Reviewer 3 Report

I think the work is solid, but it still needs to address major shortcomings.

Comments on the contents.

1.      When dealing with multiple RFID protocols simultaneously, it is crucial to highlight the strengths and weaknesses of each protocol. Otherwise, sentences like the line “These include… long-term management [4][5][6]” in Section I, that are referred to UHF RFID, can seem to concern shorter-range protocols like the NFC ones. The “Introduction” section should be enriched with references investigating the interplay between the different RFID technologies, like the following examples.

a.      A. Romputtal and C. Phongcharoenpanich, "IoT-Linked Integrated NFC and Dual Band UHF/2.45 GHz RFID Reader Antenna Scheme," in IEEE Access, vol. 7, pp. 177832-177843, 2019, doi: 10.1109/ACCESS.2019.2958257.

b.      G. M. Bianco et al., "UHF RFID and NFC Point-of-Care—Architecture, Security, and Implementation," in IEEE Journal of Radio Frequency Identification, vol. 7, pp. 301-309, 2023, doi: 10.1109/JRFID.2023.3268422.

c.      F. Paredes, I. Cairò, S. Zuffanelli, G. Zamora, J. Bonache, and F. Martin, “Compact design of UHF RFID and NFC antennas for mobile phones,” in IET Microwaves, Antennas & Propagation, Vol. 11, no. 7, pp. 1016-1019.

The authors could either expand the introduction or add a brief “Related Works” section to introduce the reader to such critical differences between the main RFID technologies.

2.      Figure 1 does not add any information that cannot be summarised in a brief sentence. It should be expanded with additional details or removed.

3.      The connection between several blocks introduced in Section 4 needs to be clarified. For instance, RFID AFE and the actions SEND/RECEIVE should be connected to the RFID-READER block properly. The crucial blocks should be highlighted in Figure 7 (Where is the AFE?).

4.      Romboic blocks in the flow chart of Figure 6 must report what actions correspond to the “yes” and “no” answers. Besides, “weather” has to be corrected with “whether”.

Presentation comments.

1.      Severe typos to be corrected are still present (e.g., the capitalisation after the point “This paper… the communications. the premise…”, section 2.2, first sentence; tag recognition layer two layer structure…). Further proofreading is necessary.

2.      Please format Table I and the subsection better (capitalization, row/columns in bold, etc.)

3.      Acronym capitalisation is inconsistent throughout the paper (e.g. SoC, SOC).

4.      A section summarising all the acronyms used should be added for the reader’s convenience.

Major proofreading is needed; some sections of the papers can be hard to follow.

Round 2

Reviewer 1 Report

Paper is now accepted.

English Language is now correct.

Author Response

Thanks for your help.

Reviewer 3 Report

The authors successfully improved the manuscript according to the reviewers´ indications. Only minor changes are needed.

1- The sentence "immunity to environmental noise and electromagnetic interference (EMI)" overstates UHF noise rejection capabilities. Please change that with something like "limited immunity to..."

2-  The sentence "So compared with other frequency bands of RFID tags, high-frequency tags in a wide range of applications, the largest production volume, manufacturers are also the most active" should be rewritten for the sake of readability

3- Row 1 and row 2 of Table 1 are inverted. The NFC carrier frequency is 13.56 MHz. Please, correct.
